

*RESEARCH ARTICLE*

**Title**

**Global warming alters mercury accumulation in trees**

**Authors**

Alexander Land[1][#], Aleta Neugebauer[1], Jürgen Franzaring[2], Petra Schmidt[3], Harald Biester[3][*][#]

**Affiliations**

[1] Institute of Biology (190a), University of Hohenheim, Stuttgart, Germany

[2] Institute of Landscape and Plant Ecology (320), University of Hohenheim, Stuttgart, Germany

[3] Institut für Geoökologie, Technische Universität Braunschweig, Environmental Geochemistry Group, Braunschweig, Germany

[#] These authors contributed equally to this work.

**Corresponding author**

[*] Harald Biester, h.biester@tu-braunschweig.de, +49 531 391-7240

**Abstract**

Stomatal uptake of mercury (Hg) by trees is the major pathway of atmospheric Hg to the terrestrial environment and most studies have suggested that Hg concentrations in tree rings are determined primarily by gaseous elemental mercury (GEM) concentrations in the atmosphere. Most studies on tree-ring Hg records were aiming to reconstruct historical atmospheric Hg emissions at polluted sites. However, the role of changing climate such as rising temperature and frequent drought events on tree-ring Hg concentrations has been rarely addressed. We show that the overall Hg load in tree rings in oaks and Douglas fir from contaminated and uncontaminated sites in Germany has been strongly determined by local atmospheric Hg emissions, but its long-term evolution has been largely determined by hydroclimate and did not follow the trend of GEM emissions in Europe. Due to different physiological heat adaptation strategies Central European oaks show continuously increasing tree-ring Hg concentration with rising temperature and precipitation rates during the past century, whereas coniferous trees show a strongly declining trend in Hg concentration in the same period,



which was also observed at other sites within the Northern Hemisphere. Our findings indicate that besides atmospheric Hg concentrations, climate change alters the Hg accumulation in different forest types and most likely the related transfer of atmospheric Hg to the soil.

**Keywords**

Tree-ring mercury, climate, sessile oak, Douglas fir, Northern Hemisphere tree-ring mercury records,

**1 Introduction**

Stomatal mercury (Hg) uptake of atmospheric Hg has been shown to be a major component in the global biogeochemical Hg cycle and litterfall is the major vector of atmospheric Hg fluxes to soils. Based on the assumption that tree-ring Hg concentrations reflect atmospheric Hg concentrations, tree-ring records have been used to reconstruct atmospheric pollution. At present, it is believed that stomatal Hg uptake and transport through the conductive tissue is the major pathway of atmospheric Hg into tree rings and that Hg uptake through roots or the tree's bark is of minor importance (Liu et al., 2024). However, it is widely unknown to which extent other factors than atmospheric Hg concentrations such as temperature, precipitation or physiological processes affect the Hg accumulation in tree rings. Previous studies suggest that coniferous trees are more suitable to reflect changes in atmospheric Hg loads by tree-ring Hg concentrations, whereas processes influencing the formation of the Hg signal in tree rings of broadleaved trees are poorly constrained. Pine and Larch are the best investigated tree species regarding historical Hg records. At Hg contaminated sites these tree species show close response to elevated atmospheric Hg concentration in their tree rings caused by anthropogenic or natural emissions (Gustin et al., 2022b; Scanlon et al., 2020; Peckham et al., 2019; Navrátil et al., 2018; Nováková et al., 2022; Peng et al., 2024; Kang et al., 2022; McLagan et al., 2022). Up to now most studies on tree-ring Hg records have mostly neglected the influence of changing climatic conditions, specifically the strong increase in temperature and severe drought events in the past decades. This might be attributed to the far higher tree-ring Hg concentrations in polluted trees compared to trees from unpolluted areas which may overwrite climate induced patterns. Moreover, tree-ring Hg records from different locations on hemispheric or global scale have only been rarely compared to decipher common patterns in Hg accumulation. Global climate change affects Hg loads in soils through litterfall and Hg reemission (Guo et al., 2024; Wang et al., 2019) and adjacent ecosystems (Li et al., 2020; Zhang et al., 2016) and thus are important drivers for Hg accumulation in tree rings. Here we present tree-ring Hg records of oak and Douglas fir from contaminated and uncontaminated sites (soil) covering the past ~150 years to decipher climatic and soil Hg related controlling factors of Hg accumulation in tree rings. We analyzed tree-ring Hg concentrations in these two species growing in Rhineland-Palatinate, western Germany

(Fig. 1). In addition, we compared literature data of tree-ring Hg records from different sites in the Northern Hemisphere and compared them to climatic variables and the evolution of atmospheric Hg emissions.

## 2 Methods

### *2.1 Sampling sites*

Oak tree-ring samples have been taken in a mixed deciduous forest at Stahlberg. The site is located at a former Hg mining area where cinnabar ore (HgS) has been extracted at several smaller spots in the forest since medieval age and especially during 1934–42 CE, but was not processed to obtain elemental Hg at Stahlberg. Ore roasting and related atmospheric Hg emissions took place at the Moschellandsberg site approx. 6 km far from Stahlberg. A deciduous forest at Leimen about 40 km distant from the Stahlberg site was selected as an uncontaminated reference site for oak. Douglas fir has been sampled at Karlshausen an uncontaminated reference site for coniferous trees (Fig. 1).

### *2.2 Tree species and tree-ring samples*

Five trees per location at Stahlberg and Leimen (each sessile oak, *Quercus petraea* (Matt.) Liebl.) as well as at Karlshausen (Douglas fir, *Pseudotsuga menziesii var. menziesii* (Mirbel) Franco) were sampled in Rhineland-Palatinate, Germany. From each tree, a tree-ring core was taken at breast height (1 m) using an increment borer (Jim-Gem, USA). The tree rings were measured with an accuracy of 0.01 mm (Rinn, 2002-2013) and precisely dated to their specific calendar years. All single tree-ring series were cross-checked to one another to ensure accurate tree-ring dating. Then, 5-year blocks (with mid-point years 2017, 2012, 2007 and so forth) were cut with a scalpel. Additionally, the sapwood tree rings were marked and counted. The cut wood samples were stored in plastic tubes, uniquely labeled and sent to the laboratory at Technische Universität Braunschweig (Germany) for mercury concentration measurements.

From each individual tree-ring mercury (Hg) concentration time series, the values were z-score transformed to ensure equal mean and standard deviation before averaging the series to obtain mean site Hg records for Stahlberg, Leimen and Karlshausen. For further analysis the two mean site Hg records of the oak locations (Stahlberg, Leimen) were combined to a mean oak Hg record with equal weight between the two oak Hg records.

### *2.3 Mercury concentration measurements in wood samples*

The precisely dated tree-ring samples were freeze-dried to constant weight prior to Hg analyses. The Hg concentration in each milled wood sample was determined by means of cold-vapor atomic absorption spectroscopy after thermal decomposition and pre-concentration of Hg through amalgamation on a gold trap using a DMA-80 EVO III direct mercury analyzer (Milestone, USA; EPA Method 7473 (U.S. Environmental Protection Agency, (EPA), “Mercury in solids and solutions by



thermal decomposition, amalgamation, and atomic absorption spectrophotometry" (1998)). NIST
1515 (Apple Leaves, 44 ± 4 ng Hg g⁻¹,) was used as a standard reference material.

### 2.4 Regional averages of seasonal mean air temperature and precipitation sum

The climate data used were downloaded from the German Weather Service Homepage
(https://opendata.dwd.de/climate_environment/CDC/regional_averages_DE/seasonal/). We used
the regional averages of seasonal mean air temperature and precipitation sum to calculate the
spring-summer records for Rhineland-Palatinate (Germany). Spring and summer air temperature
data were averaged and spring and summer precipitation sum data were summed on annual basis
to calculate the spring-summer records. Thereafter, for each 5-year block (mid-point year of blocks
are 2017, 2012, 2007 etc.) the spring-summer temperature / precipitation sum values were averaged
and z-score transformed.

### 2.5 Multi-linear regression modeling and statistical analysis

To account for the specific proportion of spring-summer air temperature and precipitation sum
influencing the tree-ring mercury concentration, multi-linear regression modeling was performed
(Montgomery et al., 2001). We used Pearson correlation statistics to calculate associations between
tree-ring mercury concentrations and the climate records. Statistical significance was assumed at
$\alpha = 1\%$.

### 2.6 Hemisphere-wide tree-ring mercury concentration records

To analyze the synchronicity between different tree-ring Hg concentration records in the Northern
Hemisphere, we used published and available datasets of different tree species with different tree
age at the continents of North America, Asia and Europe. These Hg records were treated in the
following way: If a specific Hg record represented the mean Hg concentration of several trees, the
values were z-score transformed. In the case, when such a mean Hg record was on annual timescale,
the values were averaged for 5-year blocks with mid-point years 2017, 2012, 2007 etc. to reduce
data points for better visibility. If tree-ring Hg concentration series for several trees at a specific site
were available, a mean Hg record was calculated by averaging the Hg data by the corresponding
year or year-blocks. The tree-ring Hg records used are: Scanlon et al., 2020; Chellman et al., 2020;
Ghotra et al., 2020; Nováková et al., 2021; Wang et al., 2021; Baroni et al., 2023; Peng et al., 2024;
Fornasaro et al., 2023. Thereafter, all values were z-score transformed.

### 2.7 Global average surface temperature, atmospheric mercury concentration data

Global air temperature data were received from the National Centers for Environmental Information,
National Oceanic and Atmospheric Administration
(https://www.ncei.noaa.gov/access/monitoring/climate-at-a-glance/global/time-series).





The atmospheric mercury concentration data were received from Emissions Database for Global Atmospheric Research (EDGAR, https://edgar.jrc.ec.europa.eu/country_profile).

Leaf activity (number of days between bud break and leaf discoloration) data were obtained for the

station Otterberg (station-id: 10238, Rhineland-Palatinate, Germany) from the German Weather Service (https://opendata.dwd.de/climate_environment/CDC/observations_germany/phenology/). All values were averaged for 5-year blocks and z-score transformed.

**3 Results**

*3.1 Tree-ring mercury concentrations in oak and Douglas fir*

Measurements of mercury (Hg) accumulation in tree rings of sessile oak (*Quercus petraea* (Matt.) Liebl.) and Douglas fir (*Pseudotsuga menziesii var. menziesii* (Mirbel) Franco)) are rare and controlling factors are poorly investigated. Soil Hg concentrations at the Stahlberg site show high

heterogeneity with a median of 34 mg kg$^{-1}$ (max. 205 mg kg$^{-1}$), whereas Hg concentration in soils at the two reference sites are in the range of background values (<0.5 mg kg$^{-1}$).

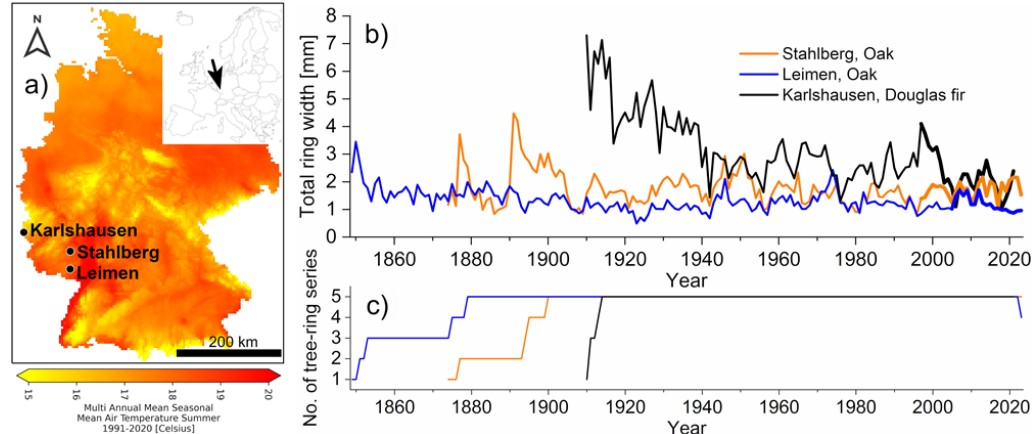


**Fig. 1:** *Area of interest and tree growth dynamic. a) Map color shows the mean air summer temperature in the reference period 1991–2020 and the geographical location of the three sites. b) Mean total ring width of oaks and Douglas firs at the different sites. c) Number of used tree-ring series available per year.*


Mean annual radial tree growth (total ring width, Fig. 1) of oaks at both sites is lower (1.8 mm at Stahlberg and 1.3 mm at Leimen) compared to Douglas fir (3.1 mm per year). The Douglas fir trees show a pronounced juvenile growth trend, which is suppressed in oak trees. The number of sapwood





rings is similar in the investigated trees (~27 in oaks at Stahlberg, ~19 in oaks at Leimen, ~25 in
Douglas fir at Karlshausen).

At the contaminated site (Stahlberg, oak) the mean tree-ring Hg concentration of 50 µg kg⁻¹ (since
1950) is 25-fold and 10-fold higher than at the uncontaminated sites at Leimen (oak) and
Karlshausen (Douglas fir) (Fig. 2, Fig. S1). Tree-ring Hg concentration variability is high among trees
at the contaminated site with concentrations ranging from 3–144 µg kg⁻¹. At the uncontaminated sites
inter-tree-ring Hg concentration variability is much more homogenous with values ranging from
<1–6 µg kg⁻¹ (oak, Leimen) and 3–11 µg kg⁻¹ (Douglas fir, Karlshausen).

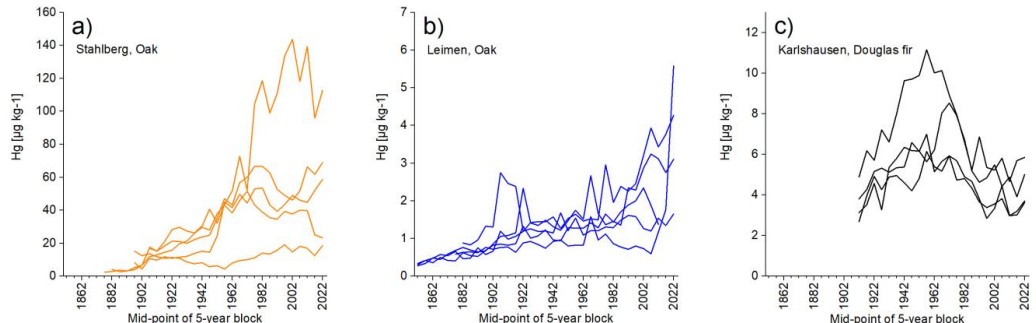

***Fig. 2:*** *Tree-ring mercury (Hg) concentration in individual trees through time from three locations in*
*Rhineland-Palatinate, Germany.*

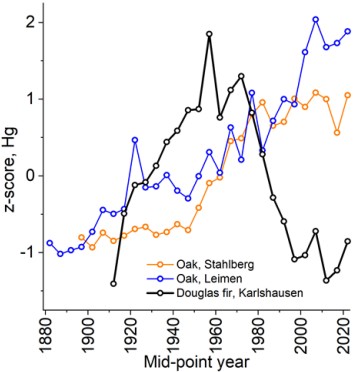

***Fig. 3:*** *Tree-ring mercury (Hg) records at the investigated oak (Stahlberg, Leimen) and Douglas fir*
*(Karlshausen) sites in Rhineland-Palatinate, Germany.*

Despite the large difference in tree-ring Hg concentrations (mean factor of 10, Fig. 2) between the
contaminated and the uncontaminated site, Hg concentrations in tree rings at both oak sites show a





clear trend of increasing values and a similar factor of increase (~9.5), although the range was higher at the Stahlberg site (2.8-26.9) than at the Leimen site (3.3-13.8).

Hg concentrations in tree rings of Douglas fir are by an average factor of two higher than in oak at the reference site, but do not show a trend of increasing values. Here, Hg concentration in Douglas fir show a maximum in the 1950s–70s and decreasing values thereafter (1980s–2000s) (Fig. 3).


## 4 Discussion

### 4.1 Differences in tree-ring Hg concentrations between sites

The high tree-ring Hg concentrations at the Stahlberg site (oak) compared to the Leimen (oak) and
Karlshausen (Douglas fir) reference sites indicate a strong local signal of high atmospheric Hg levels. We suggest that elevated Hg degassing from the Hg contaminated soil at the Stahlberg site, attributed to the widespread occurrence and high concentrations of cinnabar (HgS) and soil organic matter bound Hg in these soils (Siebelt, 2024) is the main reason for the high tree-ring Hg concentrations. This signal seems to be local only as the Hg emissions from the roasting site in
Moschellandsberg (6 km away) during World War II are not recorded in the oak trees from the Stahlberg site.

The differing absolute Hg concentrations at the two oak sites suggest an influence of local atmospheric Hg concentration, especially through soil Hg degassing. When relative tree-ring Hg variability, including the high among tree variation in factors of concentration increase, is considered,
the similar trend of increasing Hg concentrations at both oak sites suggests a common underlying controlling factor different from atmospheric Hg concentrations (Fig. 3). The same assumption is made for relative tree-ring Hg variability in Douglas fir, but with a different underlying controlling factor. Most studies up to now have investigated coniferous trees and studies on Hg in oak are quite rare. Moreover, atmospheric Hg concentrations are seen as the major factor controlling Hg uptake by
trees and climatic factors have been mostly neglected. In the following chapter we will compare published tree-ring Hg records from sampling sites all over the Northern Hemisphere with the atmospheric GEM concentration in Europe.

### 4.2 Hg tree-ring records and the evolution of atmospheric Hg emissions in the Northern Hemisphere

Our Douglas fir tree-ring Hg record shows the same pattern as those of trees from most other locations in the Northern Hemisphere disregarding local Hg contamination, tree species, forest structure or geographical attributes (Fig. 4). However, as seen for the German Douglas fir site, the strong decline in tree-ring Hg concentration by 2–3 standard deviations in the past 50 years do not correspond to the evolution of global/hemisphere atmospheric Hg concentrations, showing a Hg
decrease to a much lower extent and an offset in time (Fig.4). This discrepancy suggests that a global or hemispheric underlying factor, or a combination of multiple factors, may exert a greater influence on tree-ring Hg concentrations than atmospheric Hg levels alone. The synchronous



widespread spiking of tree-ring Hg records in the ~1970s with their subsequent rapid decline until 2000 CE needs to be driven by an at least hemispherical-wide acting factor. Climate change occurs

continental-wide and changes in air temperature in particular act as a dominant tree-growth factor, which may explain this pattern.

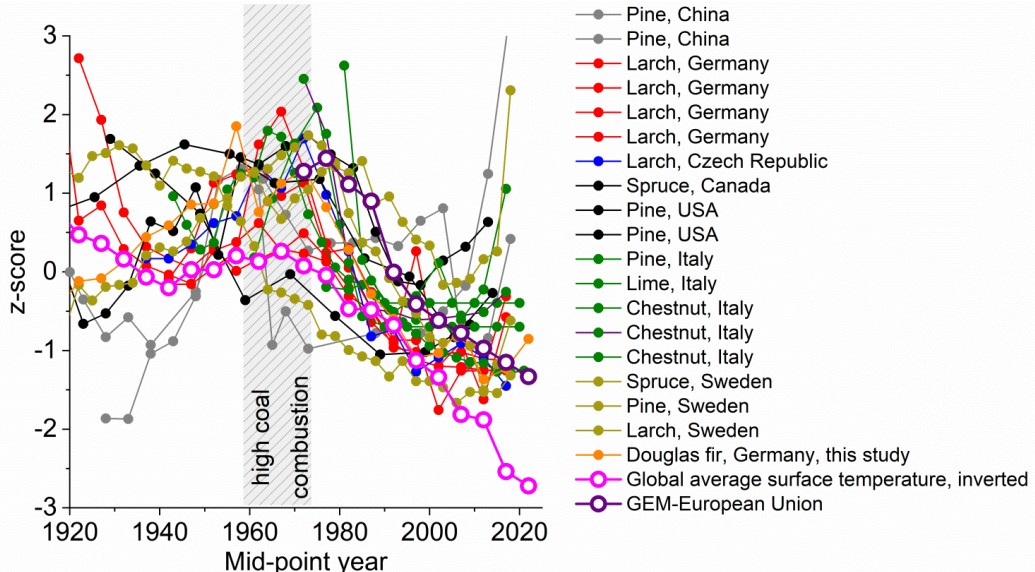

**Fig. 4:** *Records of tree-ring mercury concentration (filled points) at different sites in the Northern*
*Hemisphere (colors refer to countries)* Scanlon et al., 2020; Chellman et al., 2020; Ghotra et al., 2020; Nováková et al., 2021; Wang et al., 2021; Baroni et al., 2023; Peng et al., 2024; Fornasaro et al., 2023 *and records (circles) of global average surface temperature (inverted) and GEM emissions emitted in the European Union.*


Like in previous studies, we assume that stomatal uptake of Hg is the major pathway of Hg into tree rings and that Hg uptake by roots or bark can be neglected and radial translocation of Hg within trees (Arnold et al., 2018; Chellman et al., 2020; Gustin et al., 2022a; Liu et al., 2024; Nováková et al., 2021) can hardly explain our tree-ring Hg records. Moreover, neither our German tree-ring Hg

records nor other published European tree-ring records (Fig. 4) match with the modeled Hg emission record for Europe since 1850 (Streets et al., 2017) or emission based modeled atmospheric Hg concentration (Horowitz et al., 2014). Although both, the modeled atmospheric Hg concentration curve and our oak tree-ring records show a steady increase starting at the end of the 19th century, the first increase in modeled global, hemispheric atmospheric Hg concentrations starting from the

1850 until about 1900 as well as the second strong increase between 1930 and 1940 are almost absent in our oak tree-ring records. The peak of atmospheric Hg concentration in the 1960s–70s is poorly visible and only in some trees at the contaminated Stahlberg site, but not at the




uncontaminated Leimen site. Here, the highest Hg concentrations are reached significantly later between 2002 and 2010. The transfer function between atmospheric Hg concentrations, the Hg content in leaves and finally in tree rings is not known. It is important to note that modeled global atmospheric Hg concentration increase by a factor of about seven between 1850 and 1970 (European emission increase 4-fold), whereas concentrations in oak tree rings increase by factors of 2 to 17 and 2 to 6 at the Stahlberg and Leimen site. Hg in Douglas fir tree rings show maximum Hg concentration between the 1955 and 1970 and all other trees show a decline after 1970 by factors between 1.5 and 2 to the Hg levels of the early 20th century. This differs from modeled atmospheric Hg concentrations which show a much weaker decrease of -1.5% and -2.2% per year in the past three decades (Obrist et al., 2018). Similar inconsistency of tree-ring records with hemispheric or global atmospheric Hg concentration or Hg emission records have been reported in previous studies and were explained mainly by local Hg emission sources (Scanlon et al., 2020). The strong deviation between the tree-ring Hg data and atmospheric Hg records, along with the considerable variability observed both within and among our study sites and different tree species, indicates that additional local or regional factors—such as temperature, precipitation, air humidity, soil Hg emissions, and the physiological responses of different tree species to these factors—play a critical role in governing the accumulation of Hg in tree rings.

### 4.3 The influence of climate on Hg accumulation in oak and Douglas fir tree rings

Previous studies have shown that the stomatal uptake of Hg by trees has a physiological component which is controlled by seasonality (Jiskra et al., 2018) and recent studies have further argued that besides atmospheric Hg concentration tree physiology may have an effect on the tree-ring Hg record (Peng et al., 2024). Temperature and humidity may favor or impede tree growth and rising atmospheric $CO_2$ concentrations will in addition favor plant growth and thus the leaf surface/stomata density and the Hg uptake by leaves. Zhang et al. (2016) projected that by 2050 the annual mean GEM dry deposition flux over land will increase by 20% in northern mid-latitudes, driven by a combination of increased atmospheric gaseous elemental Hg exposure and increased vegetation and foliage density induced by $CO_2$ fertilization.

To investigate the impact of climatic parameters on oak and Douglas fir tree-ring Hg records from this study as well as regional spring and summer air temperature and precipitation data were used. The results of these climate-response analyses differ between the tree species (Fig. 5). Hg in oak tree rings is significantly increasing with rising spring-summer air temperature and spring-summer precipitation sum. Spring-summer temperature influences tree-ring Hg roughly by two thirds and precipitation sum by one third (for exact weights see formula in Fig. 5). By weighting these climatic parameters, the tree-ring Hg values can be estimated. The original and estimated oak tree-ring Hg records are significantly related ($R^2$ = 0.67, p < 0.01). However, the estimation is slightly off during



the 1970s, which suffered from an extremely hot and extraordinarily dry year in 1976 and assumingly
pronounced effects of air pollution (e.g. Hg emissions from coal burning).

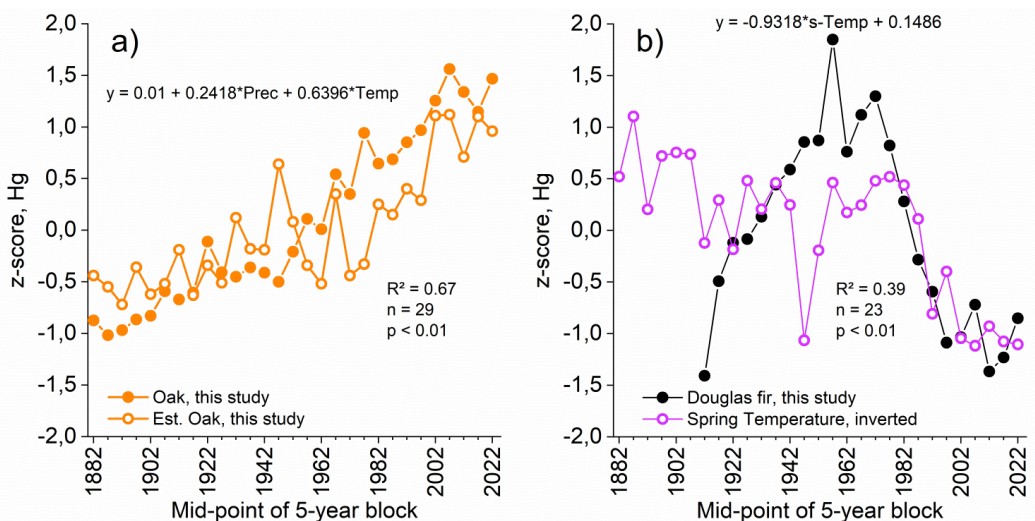


**Fig. 5:** *Climate response of tree-ring mercury (Hg) concentration variability for oak (a) and Douglas fir (b). Tree-ring Hg concentration in oak trees can be estimated by spring-summer mean air temperature (Temp) and spring-summer precipitation sum (Prec) and by spring mean air temperature (s-Temp) for Douglas fir trees (b).*


On the other hand, the decreasing Hg concentrations in Douglas fir tree rings are likely related to climate change as well. Here, the concentration of Hg is decreasing with rising spring temperature and is thus showing a statistically significant negative relationship ($R^2$ = 0.39, $p < 0.01$). The patterns of Douglas fir tree-ring Hg and spring temperature are much closer related during the past 100 years
and almost identical since the past 40 years. The extraordinary warm decade (1945–1954) after World War II did not result in lower Douglas fir tree-ring Hg, as should be expected by the before mentioned negative relationship. Instead, the Douglas fir tree-ring Hg record is peaking around this decade likely due to the distinct Hg emissions during 1950s–1970s (coal combustion). Accordingly, our results show, that Hg concentrations in tree rings are in its magnitude strongly influenced by local
anthropogenic or natural emission sources, including temperature dependent high Hg degassing fluxes from soil. In contrast, site-climatic factors, namely temperature and precipitation, appear to significantly control the Hg accumulation in oak and Douglas fir tree rings on the long term. A combination of high temperature and precipitation favor Hg uptake in oak whereas high temperature impedes it in Douglas fir. In the case of coniferous tree species, such a pronounced climate-Hg
response was previously reported by others (Zhang et al., 1995; Eccles et al., 2020). Gustin et al.



(Gustin et al., 2022b) and Liu et al. mentioned that tree-ring Hg in Ponderosa pine (California, USA) and Masson pine (Chongqing, China) are negatively correlated with temperature (Liu et al., 2024). Our data from Douglas fir supports the prevailing assumption that Hg uptake by foliage through stomata is lower when trees are under drought stress caused by rising temperature or low precipitation. For example, high temperature leads to nearly complete stomatal closure and suppressed sap flow (Samuelson et al., 2019; Niessner et al., 2024; Irvine et al., 1998) blocking foliage Hg uptake and as a consequence Hg accumulation in tree rings. Rising temperature at a coniferous site should therefore result in lower tree-ring Hg concentration, which is true for the investigated Douglas firs, with exception for the past 20 years when hemisphere-wide tree-ring Hg records are compared to global average surface temperature. Here, a disconnection can be observed which could be explained by an increase of biomass in boreal and temperate forests during the past two decades mirroring the global atmospheric $CO_2$ growth rate (Yang et al., 2023; Norby et al., 2024) and in consequence leading to a net rise in tree-ring Hg accumulation in many tree species. An additional explanation of this observation could be that rising temperature prolong tree's growing season facilitating extended Hg assimilation in relatively humid seasons (i.e. spring and/or autumn).

Not only local climate conditions play a crucial role in regards of tree-ring Hg concentration, but also how trees do physiologically respond to severe drought (Tardieu and Simonneau, 1998). One strategy to deal with drought (isohydric) is to reduce stomatal conductance (and to fully close the stomata under extraordinary dry conditions) as soil water lowers and atmospheric conditions get drier, which appears to be more frequent in mid- to high-latitude conifers. A vast number of tree-ring Hg records are established from conifers, underlining our results that Hg accumulation decreases with rising temperature-induced lower stomatal conductance. Another strategy (anisohydric) is to maintain stomata open under drought stress, allowing to perform photosynthesis. This behavior is often found in broadleaved trees. We found that tree-ring Hg accumulation in oak is positively correlated to site-specific changes of temperature and precipitation (Fig. 5). This is contrary to how other investigated tree species respond (Fig. 4). The increase in tree-ring Hg with rising temperature and precipitation is likely attributed to anisohydric eco-physiological features of oaks growing in the mid-latitudes and its temperate climate. These oaks are able to maintain sap flow and continue to operate stomatal conductance under hot and dry summer conditions (Bréda et al., 1993; Kunz et al., 2016; Raftoyannis and Radoglou, 2002; Gauthey et al., 2024), leading to continuous foliage Hg assimilation and tree-ring Hg accumulation.

The mismatch between most tree-ring Hg records and the evolution of atmospheric Hg concentrations (Fig. 4) indicates that hemispheric or global trends in atmospheric Hg loads appear to be overwritten by climate change effects. Relative changes in tree-ring Hg concentration as well as long-term trends suggest to be predominately controlled by transregional and hemisphere-wide climatic factors such as temperature, controlling tree´s stomatal conductance, the amounts of needles/leaves, annual growth rates as well as forest coverage.



## 5 Conclusions

Our findings imply that climate change, specifically rising temperatures and more frequent drought events as well as increasing $CO_2$ concentrations, may have altered the role of forests in the terrestrial biogeochemical Hg cycle. Although the relationship between tree-ring and leaf Hg concentrations is not yet known, it is likely that our observed changes in tree-ring Hg concentrations in different tree species alter litterfall Hg fluxes to soil which is the most important terrestrial vector of atmospheric Hg to soil and adjacent aquatic systems (Obrist et al., 2021; Obrist et al., 2018; Zhou et al., 2021). Moreover, global warming is forcing forest management in mid-latitudes to transfer coniferous monocultural forest into mixed or broadleaf forests, which will furthermore alter Hg fluxes to soils. During the past decades Europe's forests have increased by up to 9%, with an ~even ratio between coniferous and broadleaved trees (Köhl et al., 2020; Palmero-Iniesta et al., 2021). Climate change capable of affecting future forest productivity in different forest types in temperate and boreal regions and the practice of forestry currently undergoes a transformation process (Park et al., 2014). One forest management strategy is to expand (non-)native tree species mixture to build climate-resilient forests (Bohn et al., 2018) and planting deciduous tree species could reduce climate-driven fire occurrence (Terrier et al., 2013). Native drought-tolerant oak-rich forests, or forests containing tree species with positive temperature-Hg responses in general, can serve as a distinct sink for atmospheric Hg with changing climate. Additionally, such forests are less vulnerable to large-scale fires, release of Hg into the atmosphere could be reduced. Future forest tree species composition depends on local to regional climate evolution and forest management strategies (Park et al., 2014).

Overall, this study demonstrates that most tree species in future mid- to high-latitude forest ecosystems may be less capable to accumulate atmospheric Hg under climate change.

## Data availability

Tree-ring mercury concentration data of this study (sessile oak, Douglas fir) can be received from the following repository: Doi is added after acceptance.

## Author contributions

AL sampled the trees, dated and cut the tree rings, performed statistical analysis, interpreted the data, discussed the results and wrote the manuscript. AN sampled the trees, dated and cut the tree rings, discussed the results. JF interpreted the data, discussed the results. PS performed the tree-ring mercury measurements and discussed the results. HB performed the tree-ring mercury measurements, interpreted the data, discussed the results and wrote the manuscript.



**Competing interests**

The authors declare no competing interests.


**Acknowledgments**

We thank Marta Pérez Rodriguez and Matthias Beyer from Technische Universität Braunschweig for reviewing the manuscript.

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
