# Peer review of "Global warming alters mercury accumulation in trees"

_EGUsphere, 2025_

## Author Comment (AC2)

Answers to RC2

*RC2: 'Comment on egusphere-2025-2325', Anonymous Referee #2, 13 Aug 2025*

*The study by Land et al. explores how climate change affects mercury (Hg) accumulation in Quercus petraea and Pseudotsuga menziesii. Trees absorb mercury primarily through their stomata from the atmosphere, which is then stored in their annual rings. Previous studies focused on reconstructing historical mercury emissions on polluted sites, but the impact of climate change has been largely overlooked.*

Land et al. objectives were to investigate mercury concentrations in the annual rings of oak and Douglas fir trees in Germany over the past 150 years and to analyse (literature study) the influence of temperature and precipitation and the evolution of atmospheric Hg emissions on mercury accumulation in the Northern Hemisphere.

*Oak trees showed increasing Hg concentrations with rising temperatures and precipitation while Douglas fir trees showed decreasing Hg concentrations with rising temperatures. At a contaminated site, Hg concentrations in oak tree rings were 10 to 25 times higher than at uncontaminated sites. While Hg concentrations in oak trees have steadily increased over the past century, Douglas fir trees showed a peak in Hg concentrations during the 1950s–1970s, followed by a decline.*

*The authors speculate that Oak trees keep their stomata open during drought, facilitating mercury uptake (anisohydric) and that Douglas fir close their stomata during droughts, reducing mercury uptake. In sum, climate change and future forest tree species composition (e.g. more deciduous tree species in German forests) may have a strong influence on the Hg sink strength.*

*Overall, this study provides valuable insights for addressing Hg in forest ecosystems under changing climate conditions. However, I have some major concerns regarding the study design. The authors took only tree ring samples from 5 Oak trees from one uncontaminated and from one contaminated site (10 in total) and from 5 Douglas fir trees from another site. What kind of design is that? Were the trees randomly selected? I assume the 5 trees per site are always pseudoreplicates. Hence, the whole story remains very speculative. Additionally, I don't understand why the title is about "Global Hg concentrations" when the authors only investigated three sites with only few pseudoreplicates in Germany and the literature-study part is about the Northern hemisphere.*

Land et al.: Using a replication of 3-5 trees per site is also used in many tree-ring Hg studies. The trees were chosen randomly out of the biggest and oldest trees from each site. This is also a standard procedure when analysing Hg in tree rings as well as in dendroclimatology.

*There is no additional information about the forest sites. Was it always the same soil? What was the stocking density, age class distribution etc..? If discussing/speculating about the physiological strategies of forest tree species at different sites, one need more information about soils (soil texture, soil depth, rooting depth etc.). I guess the oak tree species were growing in more stagnic (and much more clay) environments in contrast to the coniferous tree species. How does that effects Hg accumulation in tree stems?*

Land et al.: Here we can add more information about the three forest sites. But we do not understand, which hypothesis the reviewer pursues. If root-uptake does not play a role, soil composition is also not important.

*The study is quite interesting but there is so much speculation caused by the limited sampling design. Measurements of more Oak and Douglas fir trees with comparable site conditions and/or comparing different site conditions would provide more evidence.*

Land et al.: Due to the complex tree-atmosphere-Hg interaction and the contrasting results in the literature, we have to moderately speculate about the different pathways and about there reasons. We reject the comment that our sampling design is limited. We applied standard methods and used sufficient tree series replication. Additionally, we invested more trees per species than in other publications.

*The authors should make sure not to repeat results in the discussion section and/or only discuss other literature. (e.g. The sectio4.3 The influence of climate on Hg accumulation in oak and Douglas fir tree rings" Line 282 to Line 291 is not a discussion of own results but a nice description of other literature (belongs to Introduction) and Line 292 to 299 is repetition of results.)*

Land et al.: Will be changed.

Thanks for your response.

Yours sincerely,

Alexander Land / Harald Biester

---

## Author Comment (AC3)

Dear Editor,

Thanks for sending us two detailed reviews with critical comments on our manuscript.

The reviewer comments are useful and help to improve the manuscript. Both reviewers highlight that our Hg tree ring dataset is valuable to gain insight into how climate could affect mercury tree ring concentration in oak and Douglas fir trees. However, they do not recommend publication. We offer to reinterpret and to reconsider our results. We think, that a more descriptive way of our approach would help to make the manuscript more valuable for readers.

Additionally, the title could be changed in: Climate warming may alter mercury accumulation in trees.

Please let us know, if you follow our argumentation. If our manuscript will be considered for publication, we adjust our manuscript accordingly.

Please find below our responses to the raised concerns:

*RC1: 'Comment on egusphere-2025-2325', Anonymous Referee #1, 18 Jun 2025*

*The submitted paper by Land et al. is well written, both interesting and controversial study. After reading the whole paper I was not convinced that the oak tree ring Hg records are a valid archive of atmospheric Hg pollution. Also, I believe that drawing a global conclusion about changes in the assimilation of mercury (Hg) by trees due to climate change, as indicated by the title of the paper, requires more than a study of two sites (one contaminated and one background). Many of the presented observations lack sufficient evidence and previous study including two oak species by Gustin et al. (2022) concluded that broadleaved species are not suitable for studying Hg trends in tree rings over time. But let's discuss this with authors…*

*First, increasing air mercury (Hg) due to Hg re-emission originating from increasing air temperature due to climate change is definitely plausible. The question is to what extent climatic variables affect Hg assimilation and tree ring Hg. A recent paper by Boonen et al. (2025) that was not included in the list of references assessed that 7% of the variability in the annual tree-ring Hg of Douglas fir could be explained by changes in precipitation during the active growth period. For European larch, 2% of the variability in tree-ring Hg was affected by spring temperatures. Since temperature has increased in the study area over time, a consistent trend of increasing or decreasing Hg in tree rings would correlate with temperature. However, this does not necessarily imply a causal relationship. It is implausible that a temperature increase of about 2°C would cause a tenfold increase in mercury concentrations in oak trees.*

Land et al.: Contrary to your opinion, we analysed Hg data from more than two sites. In our investigation we analysed Hg data from several locations throughout the northern hemisphere. In the paper of Gustin et al. it remains unclear why broadleaves are considered as a group of trees not suitable as historical Hg records. At the contaminated site atmospheric temperature increase resulted in a sixfold increase of Hg in oak tree rings during the past 100 years. Physiological processes in trees are often not linear but exponential. So, we do not find this implausible at the contaminated oak site. At the uncontaminated oak site the increase of tree ring Hg is threefold during the past 100 years and is, in our opinion, also realistic.

We suggest to soften the title somewhat and present our results in a more descriptive manner. We want to make clear, that our oak Hg data set is one of the most comprehensive.

*I did not fully understand the explanation as to why, at the two studied oak sites, tree ring mercury (Hg) increased over time, while none of the other oak tree ring records (nor those of many other species) from the Northern Hemisphere (except those from permafrost areas) indicated elevated Hg assimilation due to climate change. Other oak tree ring records, including those from Scanlon et al. (2020), Siwik et al. (2010), and Gustin et al. (2022), all from the Northern Hemisphere, indicated either no trend in tree ring Hg concentration or reversed behaviour with respect to annual changes in air Hg.*

*Land et al.: The data of the mentioned papers, do show similar trends. The depiction in the mentioned papers is unsatisfing. Also, no further dendroclimatological methods are used and no standardization procedure is given. This leaves the reader unclear regarding trends and climate.*

*Previous studies have shown that at sites with elevated atmospheric mercury (GEM) concentrations, trees typically have elevated mercury in their assimilatory organs, resulting in elevated concentrations of mercury in their tree rings. Thus, one would expect to see reports of increasing Hg in tree tissues. However, in the German Specimen Bank (https://www.umweltprobenbank.de/de), all sampled assimilatory organs of spruce, beech, and poplar trees indicate decreasing air Hg concentrations, similar to the indications of decreasing GEM concentrations from European air monitoring. This leaves the reader with the question of why increasing Hg in the tree rings of oaks at the two studied sites (one contaminated and one background) would not be coupled with increasing air Hg. As demonstrated by McClenahen et al. (2013) in Pennsylvania, oak assimilatory organs reflect decreasing atmospheric Hg emissions and air Hg levels.*

Land et al.: Currently, we investigate Hg concentration in litter fall to get a deeper inside into the Hg concentration at the site compared to air Hg from the German Specimen Bank.

*The explanation for the increasing trend of mercury (Hg) in tree rings could be the physiological functioning of oak bole. Previous studies have shown patterns of increasing elemental concentrations in direction from the pith to the bark, as demonstrated by Lévy et al. (1996), Bukata and Kyser (2008), and Jaozandry et al. (2024). Jaozandry et al. (2024) explained increasing nutrient concentrations (and also some pollutants) from pith to bark by translocation of elements between rings. The porous character of oak wood, which is typical of vessels (diffusive-porous wood type), is thought to play a significant role in radial translocation. The authors quickly dismiss the alternative explanation that the temporal mismatch between the tree-ring Hg and Hg emissions in Fig. 4 (which seems rather small to me) is caused by radial translocation. As we know from literature in case of conifers translocation occurred the other way i.e. contrastingly to oaks in direction from bark to the pith. For broadleaved species like oak, it is much more plausible that a substantial portion of Hg (and many other nutrients and other elements) remains in the xylem sap or is re-mobilized and moves outward as the sapwood grows.*

Land et al.: First of all, Figure 4 do not show any oak tree ring Hg series. When comparing tree ring Hg series from the northern hemisphere, a clear overall trend is seen throughout different species and different age classes. If we would follow your argumentation, it would mean that different tree species with different age translocate Hg radially from bark to pith and store Hg within the same precisely dated tree rings throughout the northern hemisphere? In our opinion, this is not plausible. Thus, it is more likely that there is a close relationship between atmospheric Hg, temperature change and tree ring Hg concentration.

*At the same time, the absence of a mercury (Hg) signal in the tree rings at the cinnabar smelting site raises questions about the validity of oak tree rings as an archive of atmospheric pollution. As indicated in the beginning of this assessment, broadleaved species were evaluated as not suitable for evaluation of Hg trends in tree rings over time (Gustin et al. 2022). Although Gustin et al. (2022) paper is in the list of references, no discussion was dedicated to addressing this statement.*

Land et al.: The sampling site is about 6 km distant from the former smelter and not in the main wind direction. Moreover, the amount of ore roasted was relatively small. We believe that the relatively high Hg emission from the local naturally Hg loaded soils has overwritten the distant atmospheric signal. This is supported, by the observation, that Hg in fir needles is highly elevated today indicating a local dominant source. Gustin et al. (2022) analysed only a few oak trees from one location in their study. Gustin et al. (2022) do not postulate that oak species is not suitable for Hg studies.

*Additionally, I don't believe the distinction between isohydric and anisohydric species adequately explains the significant variations in tree-ring mercury (Hg) levels. In study of previously mentioned Boonen et al. (2025), European larch (anisohydric) produced similar results to Douglas fir (isohydric), which suggests that stomatal behavior may not be a key factor in long-term trends.*

Land et al.: To answer that, much more investigations are needed. According to our findings, we think that the isohydric / anisohydric theory is adequate to explain the differences. But we understand the reviewer´s concern. We propose to discuss this explanation as one possible explanation.

*Given the above-mentioned reservations, I cannot recommend publication of this paper unless the authors reconsider and reinterpret their observations.*

Land et al.: Given our above responses, we offer to reconsider and reinterpret our results.

More comments and remarks:

Sampling sites

*In a study dealing with climate change I would expect to have a complete information on site elevation, mean annual precipitation and temperature to have a background for further interpretation and discussion. This information is missing.*

Land et al.: We will add this information.

Paragraph 95

*The method of sapwood tree ring estimation is missing… Another issue is the specification of mid-point years 2017, 2012, 2007 and so forth… when were the trees sampled and why is the first midpoint 2017?*

*Land et al.: Sapwood tree rings were simply counted. They show a light brown colour compared to the* dark brown colour of the hardwood. We think, that this information is not important for the reader. We have chosen these mid-points, because they reflect a five year increment interval (e.g. 2015-2019). Scanlon et al. (2020) have chosen a ten year interval (e.g. 2004-2013). Siwik et al. (2010) cut 3 or five tree rings for an interval (depending on the species). Thus, we see no issue in our specification of the mid-points. Our mid-points start in 2022, 2017 is not correct and will be changed.

Paragraph 105

*Here is a major problem, why would you aggregate contaminated site and uncontaminated site tree ring Hg records for further analysis – this does not make sense at all. If the contaminated site has been affected by the local emission or re-emission sources then it should not be used for extrapolation with the regional averages. Why was the further analysis not performed on the record of background site only?*

Land et al.: For our analyses we used standard methods from dendroclimatology. This includes a standardization method to have all single tree time records the same mean value and standard deviation. This should always be done when combining several tree time series to one "mean" series (often called chronology). After analysing contaminated and uncontaminated Hg records separately, we found no difference between them. To receive a more robust "mean" series for the climate-Hg study, we decided to increase the number of tree time series. The higher the replication of a chronology the more robust it is.

Paragraph 155

*Where in paper was the leaf activity used?*

Land et al.: We do not understand this question.

Paragraph 185

*I could not find Fig. S1 within the submission…*

Land et al.: That is correct. There is no Fig. S1 included in the manuscript.

Paragraph 200

*The concentration ranges should have units.*

Land et al.: Will be added.

*Paragraph 215*

*If such an important strong Hg emission source would not be recorded in a Hg tree ring record I would end up asking myself if the tree ring record records atmospheric Hg levels or not? The possible tree physiology effects such as radial translocation are kind of left behind in this paper. Many previous works (including those in the list of refs) indicated risks of using the sapwood tree rings without caution. Example for all one of the most recent papers Peng et al. 2024.*

Land et al.: See our comments above to the topic of radial translocation. The sampling site is about 6 km distant from the former smelter and not in the main wind direction. Moreover, the amount of ore roasted was relatively small. We believe that the relatively high Hg emission from the local naturally Hg loaded soils has overwritten the distant atmospheric signal. This is supported, by the observation, that Hg in fir needles is highly elevated today indicating a local dominant source.

*Paragraph 220*

*…studies in oak are quite rare. Please include the refs so that readers can refer to these studies.*

Land et al.: Here we will add some references.

Paragraph 320-360

*This part of the discussion is missing the fact recently published by Peng et al. (2024) indicating that sapwood tree rings should be interpreted with high caution.*

Land et al.: What does „high caution" mean? In relation to what? This remains totally unclear in all papers and shows that there is still a big lack of knowledge. We think, that our manuscript opens a new door in terms of how Hg data in tree rings could also be seen and discussed.

*Paragraph 345*

*…soil water lowers… e.g. soil moisture decreases… would be better*

Land et al.: Will be changed to soil moisture decreases.

The following Refs will be added.

Refs

Boonen, K., Shetti, R., Navrátil, T., Nováková, T., Rohovec, J., & Lehejček, J. (2025). Atmospheric mercury pollution recorded in conifer tree rings: Disentangling the effects of tree-ring width, water content, and climate on mercury concentrations. Dendrochronologia, 126370.

Bukata, A. R., & Kyser, T. K. (2008). Tree-ring elemental concentrations in oak do not necessarily passively record changes in bioavailability. Science of the total environment, 390(1), 275-286.

Gustin, M. S., Ingle, B., & Dunham-Cheatham, S. M. (2022). Further investigations into the use of tree rings as archives of atmospheric mercury concentrations. Biogeochemistry, 158(2), 167-180.

Jaozandry, C. C., Leban, J. M., Legout, A., van der Heijden, G., Santenoise, P., Nourrisson, G., & Saint-André, L. (2024). Advances in assessing Ca, K, and Mn translocation in oak tree stems (Quercus spp.). Heliyon, 10(13).

Jaozandry, Caroline Christina, et al. "Advances in assessing Ca, K, and Mn translocation in oak tree stems (Quercus spp.)." Heliyon 10.13 (2024).

Lévy, G., Bréchet, C., & Becker, M. (1996). Element analysis of tree rings in pedunculate oak heartwood: an indicator of historical trends in the soil chemistry, related to atmospheric deposition. In Annales des sciences forestières (Vol. 53, No. 2-3, pp. 685-696). EDP Sciences.

McClenahen, J. R., Hutnik, R. J., & Davis, D. D. (2013). Spatial and temporal patterns of bioindicator mercury in Pennsylvania oak forest. Journal of Environmental Quality, 42(2), 305-311.

Peng, H., Zhang, X., Bishop, K., Marshall, J., Nilsson, M. B., Li, C., ... & Zhu, W. (2024). Tree Rings Mercury Controlled by Atmospheric Gaseous Elemental Mercury and Tree Physiology. Environmental Science & Technology, 58(38), 16833-16842.

Scanlon, T. M., Riscassi, A. L., Demers, J. D., Camper, T. D., Lee, T. R., & Druckenbrod, D. L. (2020). Mercury accumulation in tree rings: observed trends in quantity and isotopic composition in Shenandoah National Park, Virginia. Journal of Geophysical Research: Biogeosciences, 125(2), e2019JG005445.

Siwik, E. I., Campbell, L. M., & Mierle, G. (2010). Distribution and trends of mercury in deciduous tree cores. Environmental Pollution, 158(6), 2067-2073.